# Intercalation of small molecules into DNA in chromatin is primarily controlled by superhelical constraint

Rosevalentine Bosire[1,2], Péter Nánási Jr.[1,2], László Imre[1], Beatrix Dienes[3], Árpád Szöőr[1], Anett Mázló[4,5], Attila Kovács[6], Ralf Seidel[7], György Vámosi[1], Gábor Szabó[1]*

1 Department of Biophysics and Cell Biology, Faculty of Medicine, University of Debrecen, Debrecen, Hungary, 2 Doctoral School of Molecular Cell and Immune Biology, University of Debrecen, Debrecen, Hungary, 3 Department of Physiology, Faculty of Medicine, University of Debrecen, Debrecen, Hungary, 4 Department of Immunology, Faculty of Medicine, University of Debrecen, Debrecen, Hungary, 5 MTA-DE Cell Biology and Signalling Research Group, University of Debrecen, Debrecen, Hungary, 6 Department of Radiation Therapy, Faculty of Medicine, University of Debrecen, Debrecen, Hungary, 7 Peter Debye Institute for Soft Matter Physics, University of Leipzig, Leipzig, Germany

* szabog@med.unideb.hu

**Data Availability Statement:** All relevant data are within the paper and its Supporting Information files.

## Abstract

The restricted access of regulatory factors to their binding sites on DNA wrapped around the nucleosomes is generally interpreted in terms of molecular shielding exerted by nucleosomal structure and internucleosomal interactions. Binding of proteins to DNA often includes intercalation of hydrophobic amino acids into the DNA. To assess the role of constrained superhelicity in limiting these interactions, we studied the binding of small molecule intercalators to chromatin in close to native conditions by laser scanning cytometry. We demonstrate that the nucleosome-constrained superhelical configuration of DNA is the main barrier to intercalation. As a result, intercalating compounds are virtually excluded from the nucleosome-occupied regions of the chromatin. Binding of intercalators to extranucleosomal regions is limited to a smaller degree, in line with the existence of net supercoiling in the regions comprising linker and nucleosome free DNA. Its relaxation by inducing as few as a single nick per ~50 kb increases intercalation in the entire chromatin loop, demonstrating the possibility for long-distance effects of regulatory potential.

## Introduction

The nucleosomes interspersed with linker DNA and the linker histones form the basic repeating unit of chromatin serving, from a structural point-of-view, to compact and package eukaryotic DNA [1]. On the other hand, accessibility of DNA within the chromatin to regulatory factors controls transcription, replication, recombination and repair [2]. A combination of nuclease hypersensitivity assays with next generation sequencing (NGS) techniques have been used to characterize chromatin accessibility genome-wide. The most widely used techniques include MNase-seq [3], DNase-seq [4], ATAC-seq [5] and NOMe-seq [6]. Nuclease sensitive sites are regarded as open chromatin, accessible to regulatory factors, whereas the

**Funding:** This work was supported by: GSz GINOP-2.3.2-15-2016-00044 -https://nkfih.gov.hu/funding/otka; GSz OTKA K128770 -https://nkfih.gov.hu/funding/otka; GSz COST CA 1521 -https://nkfih.gov.hu/funding/otka; and Stipendium Hungaricum awarded by the Tempus Public foundation (https://tka.hu/english).

**Competing interests:** The authors have declared that no competing interests exist.

nucleosome covered regions are referred to as closed, and are considered inactive, due to being occluded from the regulatory factors. Transcription is enabled by the generation of nucleosome free regions (NFR) at the transcription start sites (TSS) and other regulatory regions involving the activity of chromatin remodelling enzymes recruited to these sites [7–9].

Interpretation of accessibility is not obvious, as the DNA is wound around the nucleosome, readily accessible to molecules that bind to the DNA grooves [10,11]. On the other hand, when protein binding is shielded by the nucleosomal structure, access to the DNA could be facilitated by the spontaneous transient unwrapping of DNA ends from the nucleosome surface [12]. Hence the closed state is often perceived to be as a result of higher-order packaging involving internucleosomal interactions [13], partly mediated by the acidic patch of the H2A-H2B dimers and H4K16 [14]. Linker histone binding as well as the length of the linker regions also influence the pattern of internucleosomal interactions leading to the formation of either a compact two-start helix or the more open one-start helix [15,16]. The longitudinal compaction of chromatin has been experimentally shown to vary along the chromatin fibres [17] consistent with the euchromatin-heterochromatin dichotomy. NFRs would obviously be less prone to establish internucleosomal interactions, therefore in such regions the primary and the higher-order packaging would correlate. However, also in view of their small size relative to the diffusion barriers present [18,19], these mechanisms do not readily offer an explanation for the lack of binding of transcription factors, other than the pioneer transcription factors [20], to nucleosomal DNA.

DNA shape as well as sequence motifs are recognized by DNA binding proteins including transcription factors [21–23] and high mobility group box (HMGB) proteins [24,25]. Their binding in turn causes DNA to unwind, kink and bend facilitating formation of the multi-protein assemblies required for transcription [26,27]. Intercalation of the hydrophobic amino acid residues is central to the binding, bending and kinking of DNA [28,29] and mutation of the intercalating residues has been shown to inhibit HMGB1 binding to chromatin [30]. Inhibition of intercalation by the nucleosome has also been shown for the small molecule intercalator, ethidium bromide (EBr) [31–34]. However, if intercalation into nucleosomal DNA is inhibited by a mechanism involving molecular shielding or rather the constraint of its superhelical twist, has not been clarified.

DNA conformation is primarily determined by its superhelical state which is established during chromatin assembly in S phase of the cell cycle. Following DNA replication, DNA is wrapped around the histone octamer generating constrained negative supercoils and compensatory positive supercoils on the linker DNA. On the other hand, DNA replication involves the excessive generation of negatively supercoiled DNA in the nascent leading strand [35], so the linkers and NFRs may become negatively supercoiled. Unconstrained, extranucleosomal superhelicity is continually relaxed genome-wide by topoisomerase action behind the replication fork [36]. However, whether relaxation is overall complete, has not been clearly established. Early studies characterizing *in vivo* torsion based on the preferential binding of the intercalating drug psoralen to negatively supercoiled compared to relaxed DNA, observed no change in its binding to eukaryotic cells following relaxation of DNA using x-ray or gamma irradiation [37,38]. However, in a more recent study, reduction in psoralen binding ensued following treatment of cells with bleomycin, a nicking agent, as observed by fluorescence microscopy [39]; this was interpreted in a sequel to that paper [40] to imply that the eukaryotic genome harbours a level of extranucleosomal torsion, alluded to as net superhelicity.

During transcription, the DNA is forced to rotate around its own helical axis generating one positive and one negative supercoil ahead and behind, respectively, for every 10.5 bp transcribed [41,42]. Some of the thus generated supercoils could be absorbed during the disassembly and reassembly of the octasome, [43–45], but they are also thought to be relaxed by DNA

topoisomerases [46–48]. Whether the transcription-induced changes in supercoiling are symmetrically or asymmetrically relaxed eventually on a global scale, is less clear. Domains containing either negative or positive supercoiling relative to each other and changing in a dynamic fashion upon transcriptional inhibition were detected in human chromosome 11 [39], but it is hard to tell whether these effects cancel out each other or contribute to a net superhelicity of the genome.

In view of the fact that intercalating hydrophobic amino acids contribute to the formation of many protein-DNA complexes, especially transcriptional regulators [23], exploring intercalation of small molecules may provide valuable information reflecting on this crucial aspect of their complex binding mechanism. Here we used an *in situ* assay to characterize intercalation of fluorescent dyes and psoralen into the genomic DNA in a close-to-native state of the chromatin. This allowed us to observe an unexpectedly tight control of intercalator binding by the nucleosome structure which could be explained by the constraint of the superhelical state of nucleosomal DNA. The data presented also support the notion that there is a net overall superhelicity in the extranucleosomal DNA regions and provide evidence for long-distance effects of loop relaxation.

## Results

### The cell membrane is not the only barrier to ethidium bromide intercalation *in vivo*

By incubating live HeLa cells with the intercalating dye EBr, we observed that the dye is taken up and its fluorescence is observed in the cytoplasm and nucleoli but not in the chromatin. (Fig 1A–1C). Furthermore, when the dye was directly microinjected into the cytoplasm in a whole-cell patch clamp set-up that enabled us to monitor plasma membrane integrity in the course of the experiment, staining of the chromatin was delayed for several minutes (Fig 1D–1I). This is despite the fact that the dye, after diffusing across the cytoplasm as shown by the gradual spreading of increased cytoplasmic, likely mitochondrial, fluorescence, already stained the nucleoli located distal from the point of microinjection. When the cells were permeabilized with Streptolysin O toxin, Triton X-100 or digitonin, the fluorescence of nucleoli was clearly higher than that of chromatin (S1 Fig). These observations together suggest that the chromatin exhibits considerable resistance to intercalation.

### Intercalation into nucleosomal DNA closely correlates with nucleosome core particle (NCP) disassembly

To determine the effect of the nucleosomal structure on intercalation, we analysed the amount of intercalated EBr in salt pre-treated nuclei by laser scanning cytometry (LSC). Treatment of nuclei with varying concentrations of salt leads to histone dissociation from chromatin in a concentration dependent manner [49]. To eliminate the contribution of double stranded RNA to the measured EBr fluorescence, nuclei were digested with RNase A prior to histone elution. Without RNase A treatment, EBr staining showed a biphasic dependence on the salt concentration used for pre-treatment of the nuclei (S2 Fig). For both HeLa and human peripheral blood mononuclear cells (HPBMCs) nuclei, we observed that the mean EBr fluorescence remained unchanged upon pre-treatment with up to 750 mM NaCl. Above this concentration, a gradual increase in EBr fluorescence was observed that coincided with the destabilisation of the core nucleosome particles (Fig 2A and 2B). A similar increase in binding was observed for the intercalating drug psoralen and the bis-intercalator YOYO-1, also mirroring nucleosome eviction (Fig 2C and 2D respectively). At the same time, DAPI that binds to the DNA minor

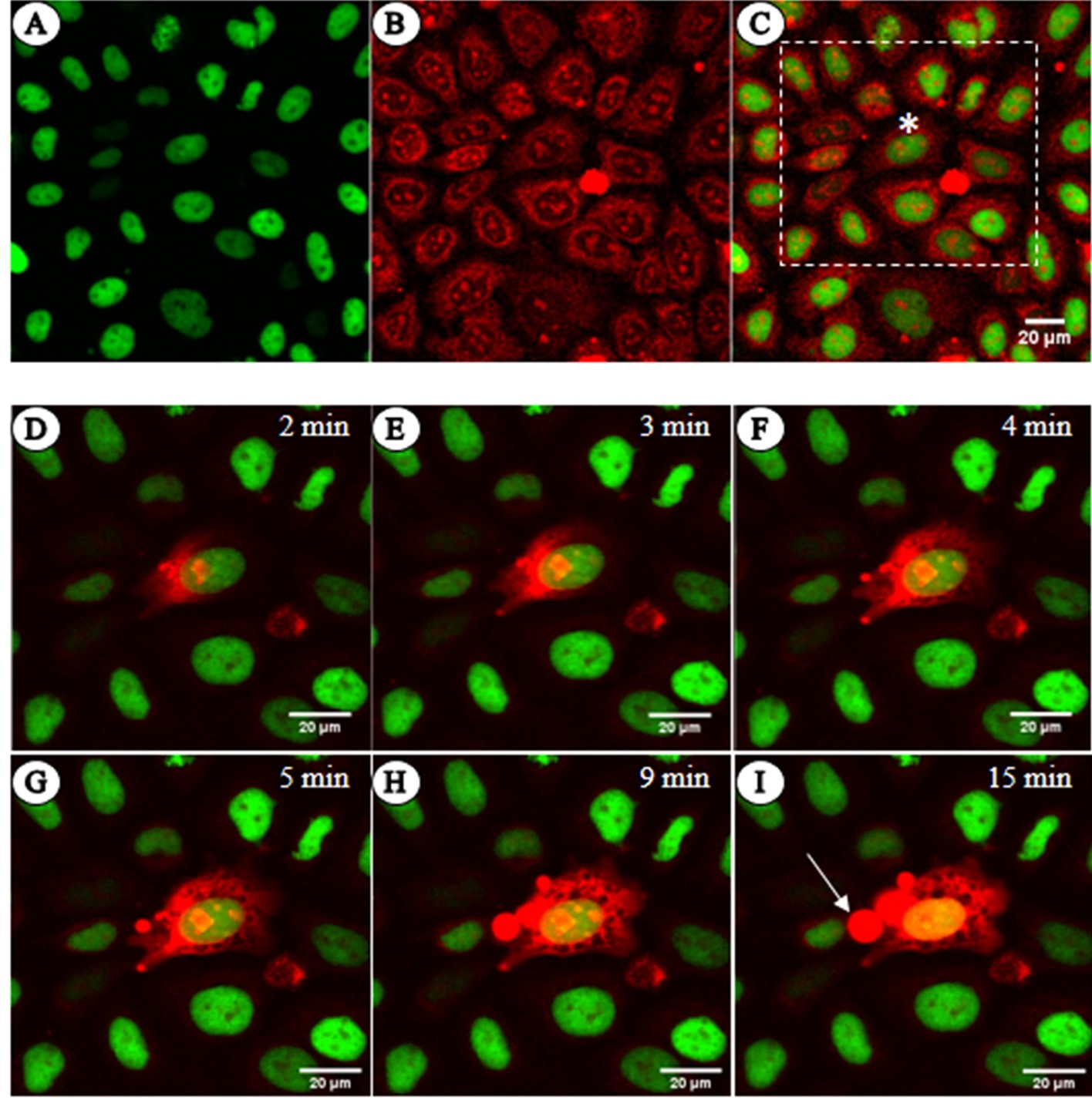

**Fig 1. Native chromatin does not readily stain with EBr.** (A-C) Live HeLa H3-GFP cells stained with EBr and imaged by CLSM. (A) H3-GFP, (B) EBr and (C) merged image. (D-I) Merged images of the cells within the white dashed square in C, taken at the indicated time points following microinjection of EBr into the cell marked by an asterisk (*). The arrow points at a cytoplasmic bleb filled with EBr likely binding to lipid micelles. The concentration of EBr in the micropipette and bathing solution was 4 μg/ml. EBr fluorescence, red; GFP fluorescence, green.

groove exhibited no increase in fluorescence upon nucleosome destabilisation (also when it was added alone, in the absence of EBr staining or GFP). We also observed that the mean area

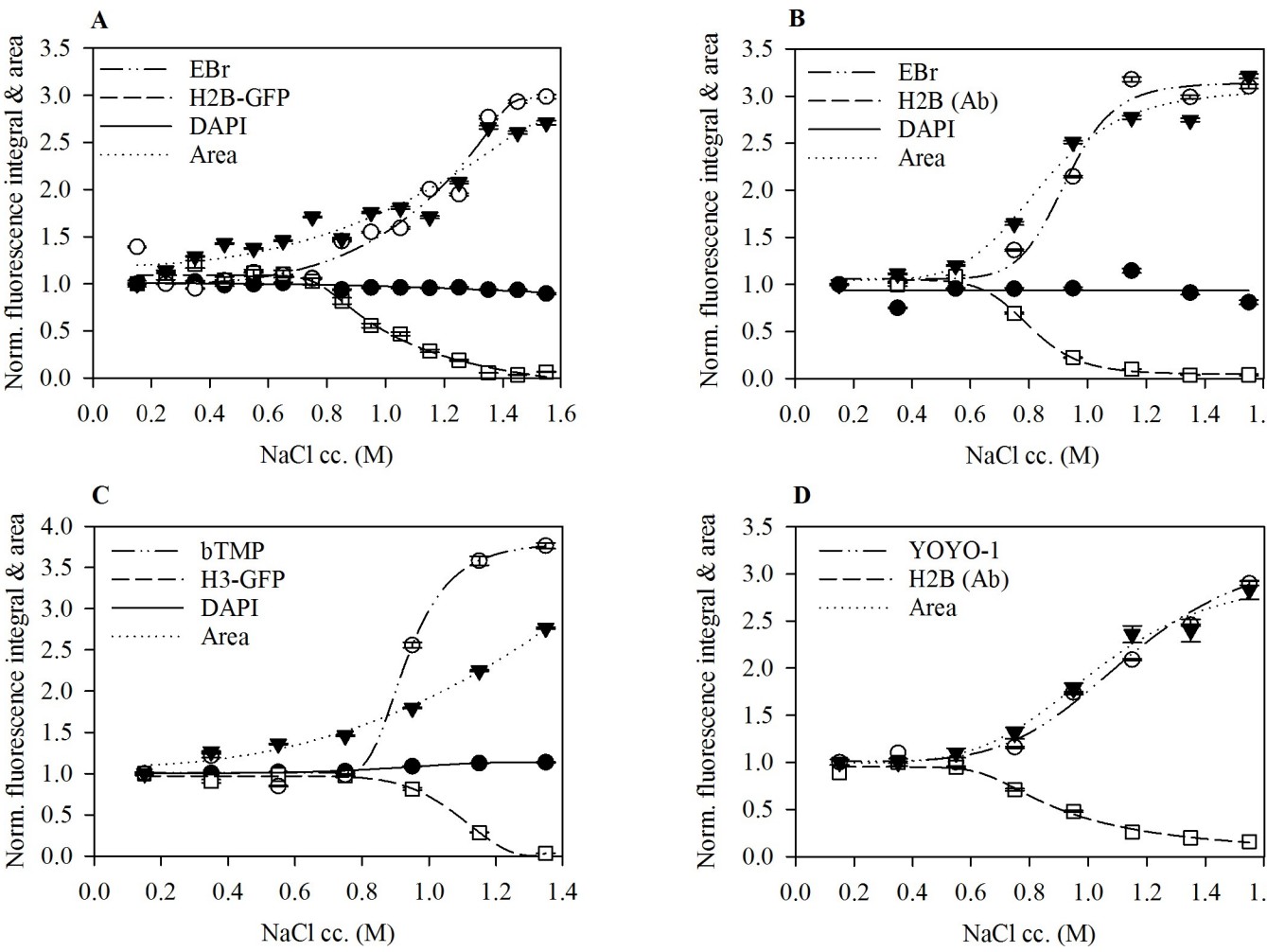

**Fig 2. Increase in intercalator binding closely correlates with core nucleosome destabilisation: Changes in area and intercalator binding following histone elution.** Histones remaining in the nuclei were detected by the GFP tag (A, C) or by immunofluorescence labelling (B, D). In each case DAPI staining remained unchanged. (A) EBr binding in HeLa H2B-GFP nuclei. (B) EBr binding in HPBMCs nuclei. (C) Biotinylated trimethylpsoralen (bTMP) binding in HeLa H3-GFP cells. (D) YOYO-1 binding in HeLa cells. Plots show one representative out of three independent experiments. Mean ±SEM of G1 phase cells normalised to control samples maintained in PBS-EDTA during the salt treatment, number of nuclei assessed n, ≥ 750.

of individual nuclei increased following salt treatment (Fig 2A–2D and S3 Fig) confirming the role of histone proteins in DNA compaction.

## Initial intercalator binding is limited to the linker and NFR regions

The initial EBr fluorescence was found to be about 30% of the maximum nuclear fluorescence obtained when all histones had been eluted, likely reflecting staining of the extranucleosomal DNA (comprising all the linker DNA and the NFRs). Consistent with this, we found this fraction to be sensitive to MNase and hypersensitive to DNase I digestion (Fig 3A and 3B respectively). Above 1 M NaCl, we observed a decrease in the number of nicks detected apparently due to loss of DNA following DNase I digestion.

It has been previously reported that EBr also binds to histones [50]. If the histones bind a sufficiently high fraction of the dye present, then in their presence the dye concentration will become a limiting factor, perhaps accounting for the increase in DNA-bound EBr after histone

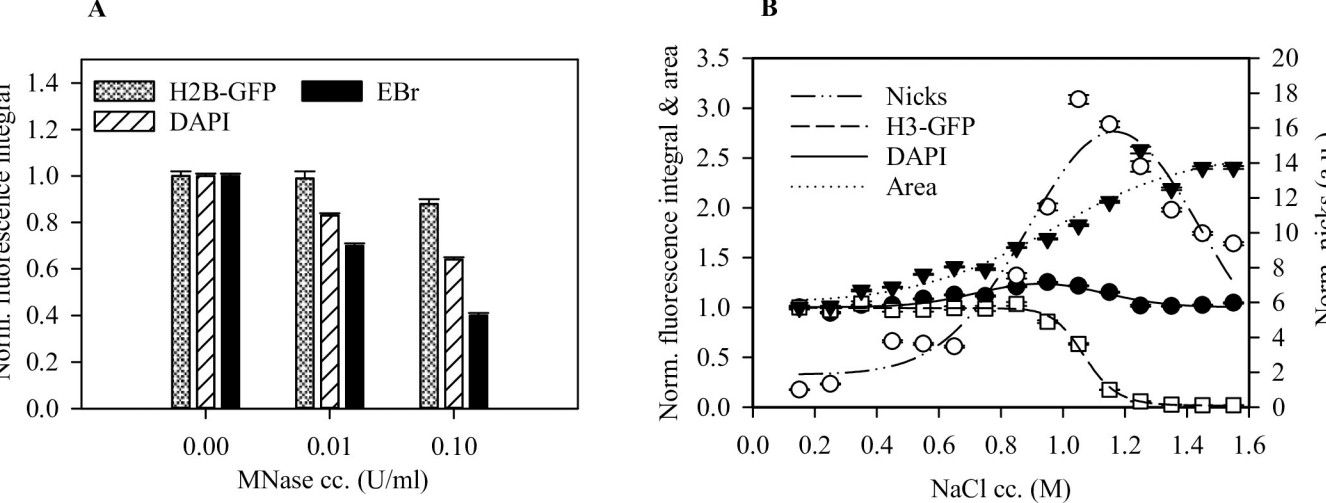

**Fig 3. Intercalator binding is limited to linker and NFR DNA in the presence of nucleosomes.** (A) Reduction of EBr staining following brief MNase digestion of formaldehyde fixed HeLa H2B-GFP nuclei. (B) Marked increase in DNase I hypersensitivity following core nucleosome destabilisation. Plots show one representative out of three independent experiments. Mean ±SEM of G1 phase cells normalised to control samples, n ≥ 750.

elution. To investigate this possibility, we determined the concentration of the dye in the supernatant after staining nuclei that were pre-treated with various concentrations of salt. Approximately 75% of the dye remained in the supernatant at all salt pre-treatment conditions confirming that the dye was not the limiting factor before histone elution (Fig 4A). Thus, the rise in EBr fluorescence accompanying nucleosome eviction reflects a parallel increase in the dye binding capacity of the genomic DNA. Indeed, fluorescence lifetime imaging of EBr in salt pre-treated, RNA depleted, EBr stained HeLa nuclei revealed a single lifetime component of 23 ns (Fig 4B) corresponding to DNA-bound EBr [51].

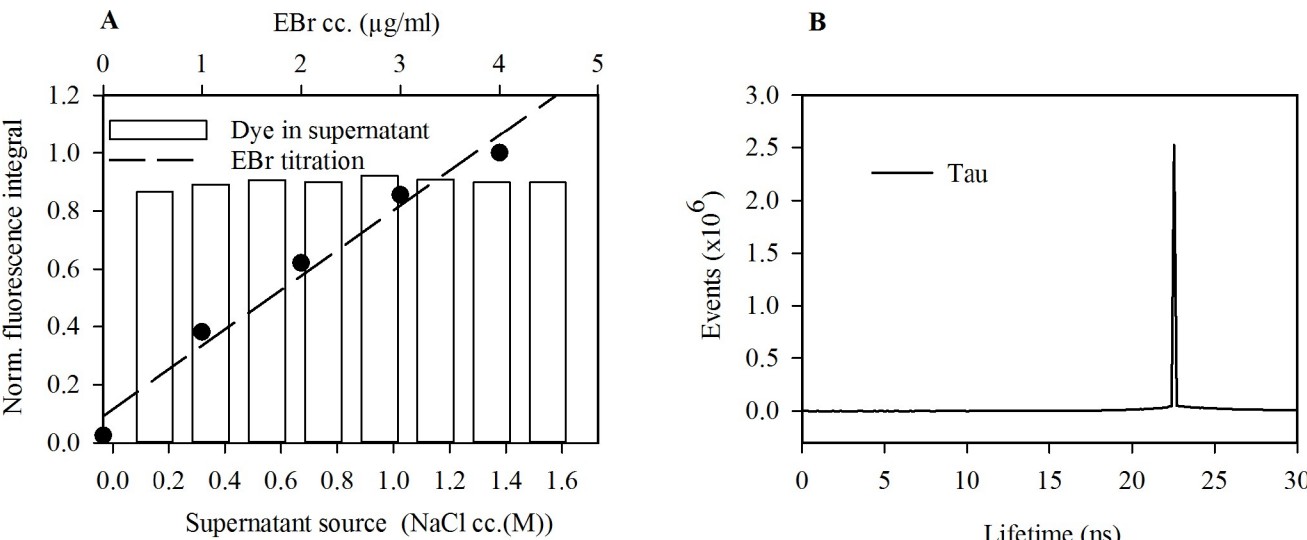

**Fig 4. EBr fluorescence reflects the amount of DNA _not_ bound to nucleosomes.** (A) The amount of dye remaining in the supernatant of the nuclei after staining. EBr fluorescence, determined by spectrofluorometry. The dashed line shows an EBr titration curve (upper X axis). (B) EBr fluorescence lifetime distribution of agarose embedded, RNA depleted HeLa nuclei pre-treated with various salt concentrations and stained with EBr. At every salt pre-treatment a single lifetime component was detected.

We also observed that EBr intercalation, but not DAPI binding, was significantly reduced if nuclei were fixed with the crosslink-forming formaldehyde prior to staining, unlike in the case of ethanol fixation that had no significant effect (S4 Fig); these observations are in line with the interpretation that topological constraint is the mechanism limiting intercalation.

### Enhancement of EBr intercalation by DNA nicking

In experiments with plasmid DNA we demonstrated that covalently closed, negatively super-coiled DNA stains less intensely with EBr compared to an equal amount of torsionally uncon-strained, nicked and linear DNA in the same gel, in the concentration range investigated (Fig 5A). This is in line with earlier data and thermodynamic calculation [52]. To determine if a topological constraint in the linker and nucleosome-free regions imposes a limitation on EBr intercalation, we exposed live cells to 300 Gy x-ray radiation, a dose sufficient to generate about 1 nick/50 kb chromatin (Fig 5B) and compared the EBr staining of the nuclei in these samples with the nuclei derived from control, non-irradiated cells. Torsional relaxation caused an increase in intercalator binding already in the presence of nucleosomes (Fig 5C and 5D). In the case of nuclei treated with high salt, i.e. in the absence of nucleosomes, there was no differ-ence in nuclear fluorescence between irradiated and control cells. *In vivo* nicking had no dis-cernible effect on the salt-induced destabilisation of H3 histones (Fig 5C) but significantly reduced the nucleosomal binding of H2B-GFP at 0.75 M salt (Fig 5D). A significant enlarge-ment of the nucleus was observed at salt concentrations where nucleosomes were still in place and the size-increment of the nuclei was augmented upon irradiation (Fig 5E).

Next, we asked if inhibition of transcription evokes any change in EBr binding. In the lower salt concentration range, we observed a small increase in the EBr staining of nuclei derived from α-amanitin or actinomycin D treated cells compared to the control cells (Fig 6A and 6B).

### Discussion

In addition to the barrier presented by the plasma membrane of viable cells, the chromatin itself resists staining with the intercalating dye EBr (Fig 1). This observation also raises the pos-sibility that stages of apoptotic cell death may exist where the membrane is already permeable but the chromatin still resists staining.

Early studies using mononucleosomes [31,32] and isolated chromatin fragments [32–34] revealed that the nucleosome structure impedes EBr intercalation into nucleosomal DNA. Whether and how this impediment is manifested *in situ*, has not been investigated in detail before. Based on more recent approaches [49,53], we have developed an assay to study intercalation with minimal perturbation to the native chromatin structure. By pre-treating agarose-embedded nuclei with a concentration series of salt, we could measure the effect of H2A-H2B dimer dissociation, occurring at ~0.75 M NaCl, and that of the (H3-H4)$_2$ tetrasome, ensuing above ̴1.2 M, on EBr intercalation. The results clearly reveal that the binding of EBr and other small molecule intercala-tors to the nucleosome-bound DNA only becomes possible after nucleosome destabilisation (Fig 2A–2D). The possibility that any binding of the intercalator by histones (as suggested by [50]) contributed to the shape of the EBr staining curves is unlikely since there was sufficient dye left in the supernatant after staining (Fig 4A) and just a single EBr component with the lifetime charac-teristic for intercalated dye [51] was present in the samples (Fig 4B). Thus, the fluorescence inten-sities measured must reflect the amount of intercalated dye only.

We have ruled out steric hindrance as a likely mechanism of inhibition of intercalation for the small molecules as binding of DAPI, which binds to the minor groove, is unaffected by the presence of histones. Thus, we hypothesize that it is intercalation, a step following access to DNA, which is inhibited by the presence of nucleosomes. This inhibition is likely due to the

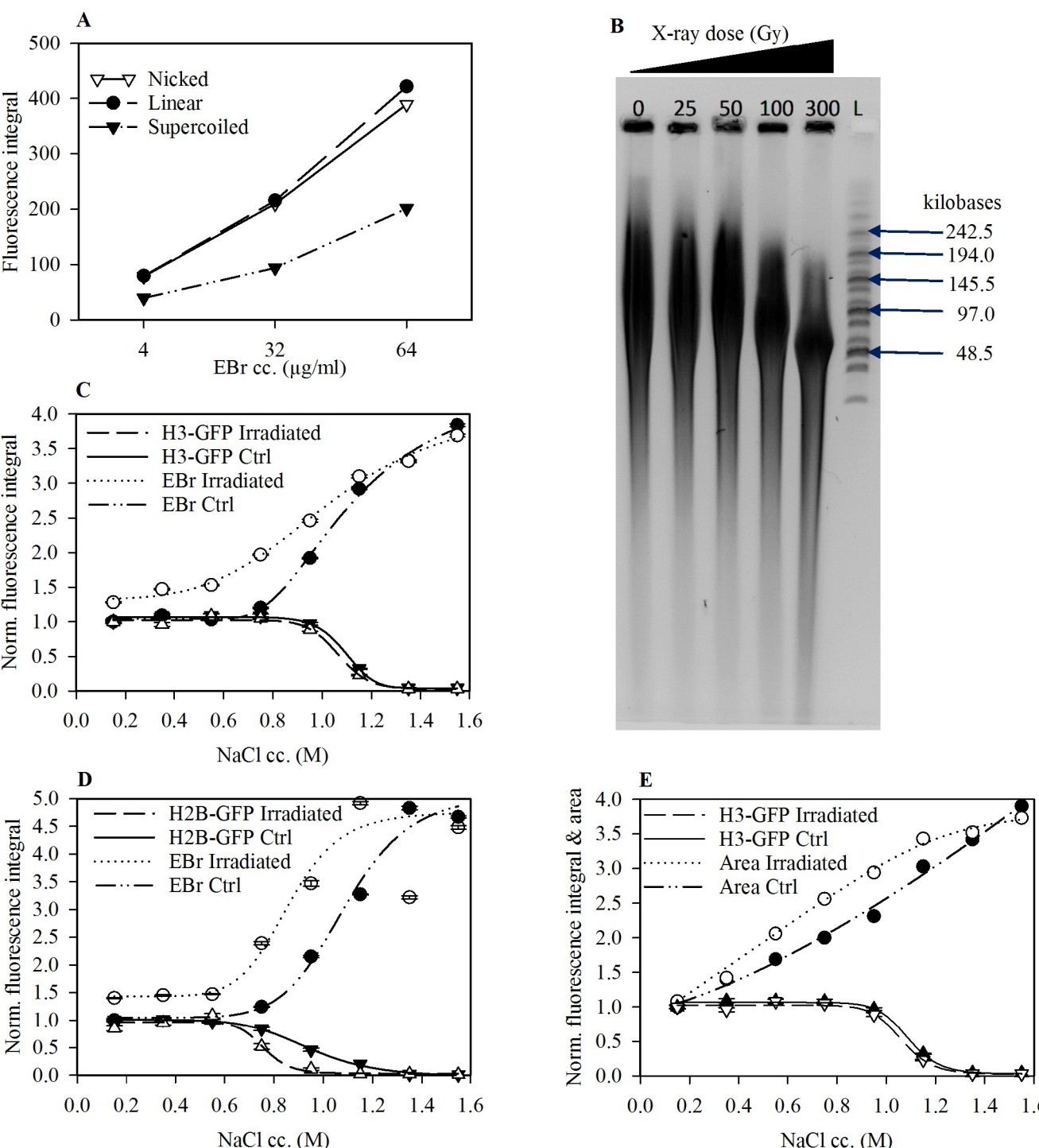

**Fig 5. Relaxation of extranucleosomal torsional tension increases EBr intercalation.** (A) EBr fluorescence profile of topologically different plasmid DNA molecules as a function of EBr concentration. (B) Nick frequency as a function of x-ray irradiation dose. The nicks were converted to double strand DNA breaks by S1 nuclease digestion. The DNA samples were analysed on agarose gels by CHEF, L; Molecular weight marker. See the original gel image in the supporting information. (C and D) EBr binding and amount of histones remaining in the salt pre-treated nuclei prepared from x-ray irradiated (300Gy) and control HeLa H3-GFP (C) and HeLa H2B-GFP (D) cells. (E) Change in area as a factor of histone elution and x-ray irradiation; same sample as C. Mean ± SEM of G1 phase cells normalised to the values obtained for control nuclei maintained in PBS-EDTA. Plot shows one representative out of three independent experiments. n≥750.

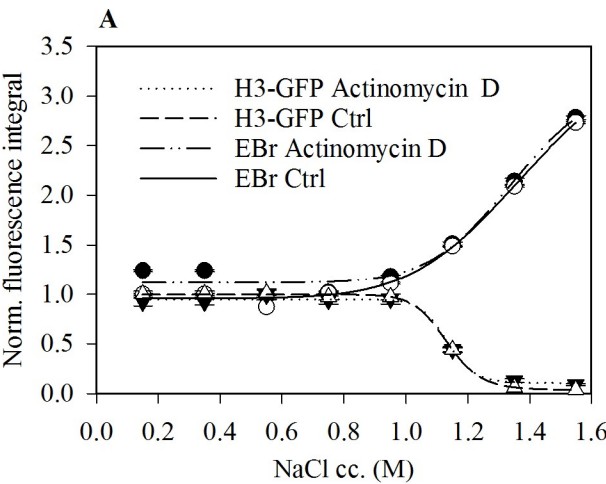
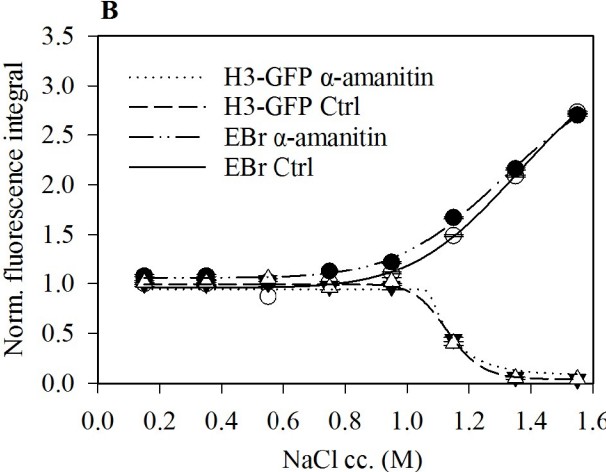

**Fig 6. Transcription inhibition causes small increase in EBr intercalation.** EBr staining of nuclei derived from HeLa H3-GFP cells treated with either 10 μg/ml Actinomycin D (A) or 50 μg/ml α- amanitin (B) compared to that of the control cells. Mean ± SEM of G1 phase cells normalised to the values obtained for control nuclei maintained in PBS-EDTA. Plot shows one representative out of three independent experiments. n≥750.

nucleosomal constraint imposed on DNA, involving hydrogen bonds and salt bridges between the phosphodiester DNA backbone and histone protein residues [54], hampering the untwisting of DNA to accommodate an intercalator. Although the nucleosomal structure is highly dynamic [55], the multiple contacts [54] collectively may significantly limit the degree of freedom of the whole DNA region wrapped around the nucleosome. This interpretation is also in line with the negative effect of formaldehyde cross-linking on EBr staining, as compared to ethanol fixation (S4 Fig). Upon salt induced nucleosome eviction from the chromatin the constrained toroidal structures are converted into plectonemes capable of changing twist and writhe to accommodate the intercalating agents.

The nucleosome core particle (NCP) organizes 147 bp of DNA in a left-handed superhelix generating a +0.2 change in the DNA helical twist ($\Delta Tw$) and a -1.5 change in writhe ($\Delta Wr$) relative to relaxed B DNA [56]. Overall, each nucleosome generates a linking number difference ($\Delta Lk$) of about –1.26 units on the average according to the general equation $\Delta Lk = \Delta Tw + \Delta Wr$ [57]. Experiments addressing the contribution of linker DNA length to the overall $\Delta Lk$ have found that the actual values range between -0.9 to -1.4 per nucleosome, with polynucleosomes of 10n repeat length having higher $\Delta Lk$ values than those with 10n+5 [15]. The difference in linker length seems to correlate with the zig-zag vs. solenoid structure of chromatin, also reflected by the heterochromatic or euchromatic partitioning, respectively [16,17]. That linker length contributes to the overall value of $\Delta Lk$ is indicative of the existence of net superhelicity outside the nucleosomes. In apparent support of such a scenario, intercalation into the linker and NFR regions of chromatin was increased as a result of DNA nicking (Fig 5C and 5D). This is unrelated to nucleosome eviction as we observed no difference in the nuclear H2B and H3 content between the irradiated and control cells below 0.75 M NaCl (Fig 5C and 5D). However, the hindered intercalation into the linker and nucleosome-free DNA of the anchored chromatin loops as well as into the covalently closed negatively supercoiled plasmid (Fig 5A) can also be explained based on topological constraint alone, i.e. without existence of net torsion (when $\Delta Lk \neq 0$). Considering that intercalation requires local DNA untwisting which in turn overwinds the adjacent DNA portions, increasing intercalation thus leads to an increasing positive supercoiling, which hinders further intercalation. On a thermodynamic level it can be described by an increased dissociation constant on positively supercoiled DNA

[58]. In case of an initial negative superhelicity which would favour intercalation compared to the relaxed DNA, the resulting positive supercoiling still strongly limits intercalation compared to the nicked molecules. The increased intercalation upon nicking of the DNA both in the plasmid (Fig 5A) and in chromatin (Fig 5C and 5D) can therefore only be taken as an evidence of topological constraint but not of a net torsion; the latter scenario is, however, supported by the arguments below.

The observed increase in nuclear area prior to nucleosome eviction (Figs 2 and 5E) suggests that the chromatin loops are held within a small volume by interactions that are more sensitive to salt than nucleosomes are. The chromatin loops probably undergo a change in *Wr*, pressing the chromatin/DNA loops against the nuclear lamina already at moderate salt concentrations. At higher salt concentrations, the chromatin loops devoid of nucleosomes project beyond the lamina through gaps in the lamina (S3 Fig). The further increase in nuclear area following X-ray irradiation (Fig 5E) already at low salt concentrations when the nucleosomes are still in place confirms that the chromatin loops are not in a relaxed state in the native chromatin, in line with the existence of net extranucleosomal superhelicity ($\Delta Lk \neq 0$).

Our findings in Fig 5C and 5D are at variance with earlier studies in which no difference in psoralen binding was observed between samples exposed to gamma or x-ray irradiation compared to control cells [37,38,59], but they are in agreement with the existence of net supercoiling observed by Naughton *et. al.* [39]. The lack of changes in psoralen binding in those early studies could be due to the low concentrations of psoralen used which would make the drug a limiting factor. In contrast, >1000x higher concentrations were used in the latter study [39]. Furthermore, psoralen forms only 1 DNA crosslink for every 15 DNA mono-adducts formed upon UVA illumination. Therefore, quantification of psoralen intercalation based on the detection of dsDNA resisting denaturation and/or exonuclease digestion [37] only captures a very small fraction of total bound psoralen, as opposed to ChIP-seq analyses and immunofluorescence detection employed by Naughton *et. al.* [39], or the detection of EBr fluorescence employed in our experiments.

The global increase of EBr binding to the linker and NFR regions following *in vivo* nicking by x-ray irradiation (Fig 5C and 5D) was observed at a nick incidence of ~ one lesion per chromatin loop (Fig 5B). These data imply that all regulatory protein binding involving intercalating moieties, e.g. of transcription factors with hydrophobic amino acids [21–23], would be simultaneously affected within a particular loop by local topoisomerase action. Therefore, our observations are particularly relevant in the context of the mechanism of transcriptional regulation by enhancers. Relaxation of topological constraints may not necessarily entail a positive effect on transcription in view of the loss DNase I and restriction endonuclease sensitivity following chromatin nicking [60,61], likely to disturb the regulation of transcriptional processes. Indeed, X-ray and gamma-ray irradiation of cells was shown to decrease, rather than increase, overall transcriptional activity [62,63].

EBr binding in the presence of nucleosomes appeared to be only slightly shifted upon transcription inhibition by either actinomycin D or α-amanitin (Fig 6). This implies that the torsional stress induced upstream and downstream of transcription [41] is equally resolved in the two directions, globally; this conclusion is not readily attained in ChIP-Seq studies [39].

DNA intercalators form a major class of DNA-targeting chemotherapeutic agents. The success of these therapies depends on their functional interaction with DNA. We have shown that relaxation of superhelicity by X-ray irradiation increases the binding of EBr to DNA (Fig 5C and 5D). This potentiation may possibly hold true for other intercalators and DNA nicking agents and also raises the possibility that treatment with anthracyclines may be facilitated by radiotherapy or chemotherapeutical regimens involving agents that cause DNA breaks, such as bleomycin, and topoisomerase inhibitors.

## Materials and methods

### Materials

All reagents were purchased from Sigma Aldrich Co. unless where otherwise stated.

### Methods

**Cell culture.**   HeLa cells expressing GFP tagged histone H2B (HeLa H2B-GFP) or H3 (HeLa H3-GFP) [64] and wild-type HeLa were cultured in DMEM and RPMI-1640 medium, respectively, both supplemented with 10% FCS, 2 mM L-glutamine, 100 µg/ml streptomycin and 100 U/ml penicillin.

**Human peripheral blood mononuclear cells (PBMCs).**   Heparinized leukocyte-enriched buffy coats were obtained from healthy blood donors drawn at the Regional Blood Center of the Hungarian National Blood Transfusion Service (Debrecen, Hungary) in accordance with the written approval of the Director of the National Blood Transfusion Service and the Regional and Institutional Research Ethical Committee of the University of Debrecen, Faculty of Medicine (Debrecen, Hungary). Written, informed consent was obtained from the blood-donors prior to blood donation, their data were processed and stored according to the directives of the European Union. PBMCs were separated from buffy coats by Ficoll-Paque Plus (Amersham Biosciences, Uppsala, Sweden) gradient centrifugation. CD14+ monocytes were positively separated from PBMCs and used immediately or incubated in RPMI-1640 media supplemented with 10% FCS (Gibco, Paisley, Scotland) and 1% antibiotic/antimycotic solution (Hyclone, South Logan, Utah).

**Live cell microscopy.**   HeLa-GFP cells cultured on 8-well ibidi slides (Ibidi, Martinsried, Germany) were rinsed 3x with PBS (150 mM NaCl, 3.3 mM KCl, 8.6 mM disodium phosphate dodecahydrate ($Na_2HPO_4.12H_2O$) and 1.69 mM potassium dihydrogen phosphate ($KH_2PO_4$)) before addition of the staining solution (4 µg/ml ethidium bromide (EBr) dissolved in PBS). Immediately after addition of the dye solution, the slide was transferred onto a pre-warmed (37°C) imaging chamber on an Olympus Fluoview 1000 confocal laser scanning microscope. Dye uptake and staining of particular cells was tracked by taking images at five minute intervals (S1 Fig). For the microinjection experiment (Fig 1), the growth medium was carefully removed and the cells rinsed 3x with PBS. In a whole-cell patch clamp set-up individual cells were microinjected with 4 µg/ml EBr while maintaining the same dye concentration in the bathing solution. The patch clamp apparatus was coupled with a Zeiss LSM 5 Live (Carl Zeiss, Oberkochen, Germany) confocal laser scanning microscope.

**Sample preparation.**   Histone elution was carried out in agarose embedded cells following the technique described by Imre *et. al.* [49]. The wells of 8-well ibidi slides were pre-coated by dispensing 150 µl of 1% (w/v) of low melting point (LMP) agarose dissolved in distilled water into each well. The unattached agarose was then quickly removed and the slide transferred onto ice for 2 min to allow the agarose stick to the bottom of the well. The slide was then transferred back onto a thermal block at 42°C for 30 min to dry the agarose. A second layer of agarose was applied by repeating the process.

50 µl of the cell suspension ($6 \times 10^6$/ml in PBS) was brought to 37°C and mixed with 150 µl of 1% (w/v) LMP agarose dissolved in PBS and also kept at 37°C. 22 µl of this mixture was dispensed to the middle of each well and immediately covered with a home-made coverslip [49]. The cells were allowed to sediment at 37°C for 2 min then the slides were transferred to ice for 2 min to allow the agarose to polymerize. The coverslips were detached from the agarose by dispensing 300 µl of ice-cold PBS into each well and then carefully removed using a hooked needle.

The samples of the agarose embedded cells were washed 3x using cold PBS then permeabilized by incubating each well with 400 μl of 1% (v/v) Triton X-100/ PBS-EDTA 2x for 10 min each time. The detergent was washed out 3x with 400 μl/well of PBS-EDTA for 3 min each time. RNA was digested using 150 μl/well of 100 μg/ml RNase A (Thermo Scientific, Waltham, Massachusetts, USA) dissolved in PBS-EDTA, for 2 hrs at 37˚C. After digestion, the nuclei were washed 3x for 3 min on ice. Histone elution by salt was accomplished by six washing steps for 10 min each using salt solutions prepared by dissolving NaCl in PBS-EDTA to yield the total amount of salt indicated on the X-axes. The salt was then washed out 3x using 400 μl/ well of cold PBS-EDTA before addition of the intercalator.

Intercalator dyes were applied at the concentration of 4 μg/ml for EBr (300 μl/well) and 40 nM for YOYO-1 (150 μl/well), for 2 hrs. The EBr concentration corresponds to ~10 μM which is near its dissociation constant (6 μM) in PBS [58] rendering dye binding highly sensitive to changes in DNA topology and twist [58]. For the psoralen experiment, biotinylated 4,5,8-tri-methylpsoralen (bTMP) (kindly provided by Nick Gilbert) was used. The samples were incubated with 1 mg/ml bTMP (150 μl/ well) for 2 hrs on ice. Unbound bTMP was removed by a brief rinse using PBS-EDTA, and the samples were exposed to UVA (365 nm) for 10 min. Samples were placed 5 cm away from the UVA lamp. This was followed by one more wash with PBS-EDTA and blocking with 1% (w/v) BSA in PBS-EDTA for 45 min. For the detection of bTMP, nuclei in each well were incubated with 1 μg/ml mouse anti-biotin antibody (150 μl/ well) overnight at 4˚C. Unbound antibody was removed in 3 washes with PBS-EDTA then the samples were incubated with 1 μg/ml Alexa 633 conjugated secondary antibody (200 μl/well; Thermo Scientific, Waltham, Massachusetts, USA) for 2hrs. Nuclei were counter-stained with 1 μg/ml DAPI and imaged using a LSC (described below).

**DNase I hypersensitivity.** Agarose embedded, RNase A digested, histone depleted nuclei were equilibrated with DNase I buffer (10 mM Tris pH 8, 0.1 mM $CaCl_2$, 2.5 mM $MgCl_2$; 400 μl/well). Digestion was accomplished by incubation with 0.1 μg/ml DNase I (300 μl/well) at 37˚C for 10 min. Residual enzyme was inhibited or removed in three washing steps with PBS-EDTA. Samples were then equilibrated with DNA polymerase I buffer (50 mM Tris-HCl, pH 7.2, and 10 mM $MgSO_4$). Nicks generated by DNase I were labelled by *in situ* nick translation for 10 min at room temperature using 24 U of DNA polymerase I, 5 nmol of dATP, dCTP, dGTP and biotin-dUTP per well, in Pol I buffer (total reaction volume was 120 μl/ well). Biotin immunolabeling and imaging was carried out as described above.

**MNase digestion.** Agarose embedded RNA-depleted nuclei were fixed with 4% formaldehyde on ice for 20 min. Samples were then equilibrated by 3 washes with MNase buffer (50 mM Tris, pH 7.5, 1 mM $CaCl_2$) before digestion for 7 min at 37˚C, using the indicated concentration of MNase (150 μl/ well). The enzymatic reaction was stopped by washing 3x with PBS-EDTA.

**EBr staining of plasmid DNA.** 0.5 μg of native plasmid DNA (pcDNA3-EGFP, 6159 bp) was nicked or linearized using 1 U of Nb.BsmI or 1 U of EcoRI, respectively, in a 20 μl reaction volume, for 1 hr at 37˚C. (Both enzymes were from Thermo Scientific, Waltham, Massachusetts, USA.) Equal amounts of supercoiled, nicked and linearized plasmid DNA were mixed and loaded into wells of a 1% agarose gel. Following standard electrophoresis, the gels were stained with EBr (dissolved in distilled water (dH2O), used at the indicated concentration, for 45 min. Gels were imaged using a FluorChem Q (Alpha Innotech) gel documentation system and intensities analysed using Fiji ImageJ.

**In-gel cell irradiation and determination of nick incidence.** Cells were embedded into agarose plugs following the method described by Varga *et al*. [65]. Briefly, HeLa H3-GFP cells were washed in PBS and re-suspended at $1.3 \times 10^7$ cells/ml. The suspension was placed at 37˚C for 3 min before being mixed with an equal volume of 1.5% LMP agarose dissolved in

PBS-EDTA also kept at 37°C. 90 µl of the mixture was distributed into CHEF moulds and allowed to solidify at 4°C for 20 min. The agarose plugs were then transferred into DMEM medium in an ibidi slide and irradiated at room temperature at doses in the range of 25–300 Gy. The samples were kept at a distance of 57.2 cm from the 6 Mv radiation source.

Following irradiation, the plugs were washed 3x for 30 min in cold PBS at 4°C then submerged in lysis buffer (0.5 M EDTA, 1% N-Lauroylsarcosine, 1 M Tris, pH 8, and 0.5 mg/ml proteinase K (Thermo Scientific, Waltham, Massachusetts, USA) at 55°C for 48 hrs with change of lysis buffer after 24 hrs. Proteinase K was inactivated using 1 mM PMSF (phenyl-methylsulfanylfluoride) for 10 min at room temperature. The plugs were then washed 3x in cold Tris-EDTA (10 mM Tris HCl, 1 mM EDTA, pH 8.0).

The plugs were then equilibrated 3x with S1 nuclease buffer (250 mM NaCl, 49.5 mM Na-acetate, 0.36% $ZnSO_4$, and 1% BSA, pH 4.4) for 10, 30 and 60 min, respectively. Nicks were converted to double strand DNA breaks using 1000 U/ml S1 nuclease (Thermo scientific, Waltham, Massachusetts, USA) for 60 min at 37°C. Double-stranded DNA fragments (3–300 kb) were separated by CHEF (Contour-clamped Homogenous Electric Field electrophoresis) in a 1% agarose gel in 0.5x TBE buffer (89 mM boric acid, 89 mM Tris base and 2mM EDTA) using a CHEF-mapper (Bio RAD). The gel was stained with 0.5 µg/ml EBr and imaged using FluorChem Q (Alpha Innotech, San Leandro, California, USA) gel documentation system. Fragment size was identified using a pulse marker (Midrange PFG marker New England Bio-labs N0342S) that was run alongside the samples.

**Salt extraction of histones** was carried out using the technique described by Shechter et al. [66]. Briefly, $10^7$ HeLa cells were resuspended in 1 ml of extraction buffer (10 mM HEPES, pH 7.7, 10 mM KCl, 1.5 mM $MgCl_2$, 0.34 M sucrose, 10% glycerol, 20x dilution protease inhibitor cocktail and 0.2% NP-40) and incubated on ice for 10 min. Nuclei was recovered by centrifugation at 6,500g for 5 min and then washed in extraction buffer without detergent. Nuclei were then lysed by incubating with Tris-EDTA for 30 min on a rotator, at low speed. Chromatin was recovered by centrifugation at 6,500g for 5 min. Histones were then extracted using 400 µl of high-salt buffer (50 mM Tris-HCl pH 8.0, 2.5 M NaCl, 0.05% NP40), by slowly rotating the samples for 30 min at 4°C. The solution was then centrifuged at 16,000g and the supernatant filtered through a 3,500 MWCO spin column (Sartorius AG, Göttingen, Germany). Samples were washed twice with PBS then quantified by measuring absorption at 280 nm.

**Microscopy.** An Olympus FlouView 1000 confocal laser scanning microscope fitted with 3 lasers was used. GFP was excited using the 488 nm laser line, the 543 nm line was used for EBr, and the 633 nm line for excitation of Alexa Fluor 633. Microscopy images were taken using the 60x oil immersion objective (NA = 1.35) analysed using Fiji ImageJ.

**Automated microscopy**. was done using an iCys Research Imaging Cytometer (Compu-Cyte Corporation, Westwood, MA). The instrument is based on an Olympus IX-71 inverted microscope equipped with four lasers, photodiodes (detecting light loss and scatter) and four photomultiplier tubes (PMTs). A 405 nm solid state laser was used for the excitation of DAPI and Hoechst, the 488 nm Argon laser line was used to excite GFP and EBr, and a 633 nm HeNe laser for Alexa 633. For each salt treatment, 1500–2000 events were measured. Data analysis was performed using the iCys 7.0 software, and graphs were prepared using SigmaPlot 14.0.

**Spectrofluorimetry.** Quantification of EBr concentration remaining in the supernatant following staining of nuclei was carried out using a Fluorolog-3 spectrofluorimeter (Horiba Jobin Vyon, Edison, NJ). EBr was excited at 300 nm and the emission fluorescence measured at 590 nm. To prepare the EBr titration curve solution concentrations ranging from 0.5–4 µg/ml were used.

**Fluorescence lifetime imaging and data analysis.** The fluorescence decay dynamics were acquired using a Nikon A1 laser scanning microscope with a PicoQuant time-correlated single photon counting module. EBr was excited by a 510 nm pulsed laser at 4 MHz repetition frequency. Laser power was 1,891 μW as measured at the back aperture of the objective. Emitted fluorescence photons were collected using a 60x water immersion objective (NA = 1.27) and directed to an avalanche photo detector (Excelitas Technologies), using a 594 long pass filter. An average of 2,000 counts were collected for every sample. Data was analysed using SymPhoTime 64 software and fitted using n-Exponential reconvolution module for 2 lifetime components. Lifetime component plots were prepared using SigmaPlot 14.

## Supporting information

**S1 Fig. Representative images and fluorescence profile of EBr stained cells/nuclei.** The plots on the right show the EBr fluorescence line scans in the direction of the white arrows in the corresponding image on the left. Yellow arrow heads point at representative nucleoli falling between the red dashed lines on the line scans. Treatments are indicated on the respective panels. EBr fluorescence, red; GFP fluorescence, green.
(TIF)

**S2 Fig. Biphasic dependence of EBr intercalation on salt.** Agarose embedded, salt pretreated HeLa-H3-GFP nuclei.
(TIF)

**S3 Fig. Effect of salt treatment on nuclear size.** Representative confocal microscopy images of salt treated agarose embedded HeLa- H3-GFP nuclei. Salt concentration is indicated in each panel EBr fluorescence, red; GFP fluorescence, green, Lamin B1 Cyan.
(TIF)

**S4 Fig. Formaldehyde crosslinking significantly reduces EBr intercalation in RNA depleted, salt treated nuclei, relative to ethanol fixation.** The columns show the normalized mean EBr fluorescence of ~750 nuclei measured by LSC.
(TIF)

**S1 Raw Image. Raw image for gel blot in Fig 5B: Nick frequency as a function of x-ray irradiation dose.** The nicks were converted to double strand DNA breaks by S1 nuclease digestion. The DNA samples were analysed on agarose gels by CHEF electrophoresis. Gel was stained with 0.5 μg/ml ethidium bromide and imaged using FluorChem Q (Alpha Innotech, San Leandro, California, USA) gel documentation system; λ-Lambda DNA, L; Molecular Weight Marker (Midrange PFG marker New England Biolabs N0342S). Lanes marked by an X were excluded from the final image in Fig 5B.
(PDF)

## Acknowledgments

The authors thank Nick Gilbert (University of Edinburgh, UK) for the kind gift of bTMP and Zsolt Bácso for LSC maintenance. RB is on study leave from the Technical University of Kenya.

## Author Contributions

**Conceptualization:** Gábor Szabó.

**Funding acquisition:** Gábor Szabó.

**Investigation:** Rosevalentine Bosire.

**Methodology:** Rosevalentine Bosire, Péter Nánási Jr., László Imre, Beatrix Dienes, Árpád Szöőr, Anett Mázló, Attila Kovács, György Vámosi.

**Supervision:** Gábor Szabó.

**Writing – original draft:** Rosevalentine Bosire.

**Writing – review & editing:** Ralf Seidel, Gábor Szabó.

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
