## [Editor Report · Decision Letter 0]

25 Aug 2019

PONE-D-19-20630

Nucleosomal and internucleosomal impediments to intercalation into DNA assessed in situ

PLOS ONE

Dear Dr. Szabó,

Please accept my apologies for the delay in handling this submission, which was due to unexpected difficulties in finding Reviewers.

Thank you for submitting your manuscript to PLOS ONE. After careful consideration, we feel that it has merit but does not fully meet PLOS ONE’s publication criteria as it currently stands. Therefore, we invite you to submit a revised version of the manuscript that addresses the points raised during the review process.

We would appreciate receiving your revised manuscript by the end of September 2019. To enhance the reproducibility of your results, we recommend that if applicable you deposit your laboratory protocols in protocols.io, where a protocol can be assigned its own identifier (DOI) such that it can be cited independently in the future. For instructions see: http://journals.plos.org/plosone/s/submission-guidelines#loc-laboratory-protocols

We look forward to receiving your revised manuscript.

with kind regards,

Ronald Hancock

Academic Editor

PLOS ONE

Journal Requirements:

Reviewers' comments:

<h2>**Reviewer 1**

**This work is about the binding of small molecule intercalators to the chromatin complex (in situ). In my opinion this paper is of great interest as it sheds light on the barriers that DNA binding proteins might encounter before they can bind to their target sites. I have no critique about the content of the paper but think that some of the text should be improved.  Specifically:**

**1. The title gives somewhat the wrong impression what this paper is about as “nucleosomal and internucleosomal impediments” sound much more microscopic than what is actually done in this study. A somewhat less specific title (e.g. chromosomal impediments) would reflect the content of the study better.**

**2. In addition, some references to more microscopic views of how DNA binding proteins can find access to their target sites would be useful to provide. I am thinking here mainly of Jonathan Widom’s site exposure mechanism. Nucleosomal DNA is in principle accessible through the temporary unspooling of wrapped DNA. Do the experiments give any hints that intercalators can also bind to nucleosomal DNA or can they not compete with the histones?**

**3. page 3, line 45: The authors write: “Interpretation of accessibility is not obvious, as the DNA is wound around the nucleosome, readily exposed to the environment.” Do the authors refer here to the site exposure? Maybe here they could cite corresponding papers.**

**4. page 4, line 73: The authors mention psoralen here for the first time, without explaining what it is. Only on page 6 it is mentioned that it is an intercalating drug.**

**5. The following lines are hard to understand when reading the manuscript for the first time (but perfectly clear when reading it the second time). Maybe passages like “harbors a level of nucleosome-unconstrained torsion” etc. could be written more clearly.**

**6. The Results section starts with “We observed that the chromatin of live cells does not readily stain with the intercalating dye EBr despite its presence in the nucleus shown by the fluorescense of the nucleoli.” When reading this sentence I found completely lost. What experiment is described here? What are we looking at and why? Again, after having read the whole paper, this sentence makes sense (especially when reading it together with the last sentence of the introduction). But for first-time readers I recommend to cut the sentence into several. First explain the experiment. Then, describe what you observe. You might also remove the last sentence of the introduction and incorporate this information into the beginning of the result section.**</h2>

Reviewer 2

The experimental work in this manuscript is state-of-the-art and comprehensive and entirely supports the important conclusions.  I have only the comment that the intercalation was done in growth medium but the imaging in PBS - can it be excluded that this change of medium affects the binding of ethidium?

Some parts of the presentation could be improved. The title does not present well what appears to be the most interesting finding, that "relaxation by inducing as few as a single nick per ~50 kb enables intercalation in the entire chromatin loop, demonstrating the possibility for long-distance effects of regulatory potential".  In fact, the short title may be more informative.

In the Discussion, it would be interesting to consider the publication:  Luchnik AN, Hisamutdinov TA, Georgiev GP. 1988. Inhibition of transcription in eukaryotic cells by X-irradiation: relation to the loss of topological constraint in closed DNA loops. Nucleic Acids Res 16:5175–5190. There, "X irradiation was found to inhibit in vivo transcription . . About one DNA single-strand break per estimated DNA loop (domain) length is sufficient to explain the effect".  That work suggests that after relaxation, DNA becomes less accessible to the transcriptional machinery, whereas the present study shows that it becomes more acessible to an incalator.

---

## [Author Response · Author response to Decision Letter 0]

25 Sep 2019

All responses to reviewers have been uploaded in the file named "Response to Reviewers"

---

## [Editor Report · Decision Letter 1]

25 Oct 2019

Intercalation of small molecules into DNA in chromatin is primarily controlled by superhelical constraint.

PONE-D-19-20630R1

Dear Dr. Szabó,

We are pleased to inform you that your manuscript has been judged scientifically suitable for publication and will be formally accepted for publication once it complies with all outstanding technical requirements. The revisions have greatly improved the clarity and depth of the manuscript.

With best regards,

Ronald Hancock

Academic Editor

PLOS ONE

---

## [Editor Report · Acceptance letter]

5 Nov 2019

PONE-D-19-20630R1 

Intercalation of small molecules into DNA in chromatin is primarily controlled by superhelical constraint. 

Dear Dr. Szabó:

I am pleased to inform you that your manuscript has been deemed suitable for publication in PLOS ONE. Congratulations! Your manuscript is now with our production department. 

With kind regards,

on behalf of

Dr. Ronald Hancock 

Academic Editor

PLOS ONE